# LinBridge: A Learnable Framework for Interpreting Nonlinear Neural Encoding Models

## Abstract

Neural encoding of artificial neural networks (ANNs) aligns the computational representations of ANNs with brain responses, providing profound insights into the neural basis underpinning information processing in the human brain. Current neural encoding studies primarily employ linear encoding models for interpretability, despite the prevalence of nonlinear neural responses. This leads to a growing interest in developing nonlinear encoding models that retain interpretability. To address this problem, we propose **LinBridge**, a learnable and flexible framework based on Jacobian analysis for interpreting nonlinear encoding models. LinBridge posits that the nonlinear mapping between ANN representations and neural responses can be factorized into a linear inherent component that approximates the complex nonlinear relationship, and a mapping bias that captures sample-selective nonlinearity. The Jacobian matrix, which reflects output change rates relative to input, enables the analysis of sample-selective mapping in nonlinear models. LinBridge employs a self-supervised learning strategy to extract both the linear inherent component and nonlinear mapping biases from the Jacobian matrices of the test set, allowing it to adapt effectively to various nonlinear encoding models. We validate the LinBridge framework in the scenario of neural visual encoding, using computational visual representations from CLIP-ViT to predict brain activity recorded via functional magnetic resonance imaging (fMRI). Our experimental results demonstrate that: 1) the linear inherent component extracted by LinBridge accurately reflects the complex mappings of nonlinear neural encoding models; 2) the sample-selective mapping bias elucidates the variability of nonlinearity across different levels of the visual processing hierarchy. This study not only introduces a novel tool for interpreting nonlinear neural encoding models but also provides novel evidence regarding the distribution of hierarchical nonlinearity within the visual cortex.

## 1 Introduction

In recent years, aligning the computational representation of artificial neural networks (ANNs) with brain activity through neural encoding models significantly advances our understanding of the neural basis underlying information processing in the human brain. Previous research primarily employs linear encoding models due to their interpretability (Naselaris et al., 2011a; Yamins & Dicarlo, 2016). However, the prevalence of nonlinear neural processes is well recognized (Naselaris et al., 2011b; Wang et al., 2023; Tang et al., 2024; Jain & Huth, 2018), which may limit the ability of linear models to fully capture the complex dynamics of brain activity. This limitation not only undermines the predictive performance of linear encoding models but also restricts their capacity to effectively interpret neural activity.

With the rapid advancement of deep neural networks, nonlinear encoding models are increasingly utilized (Zhang et al., 2019; Li et al., 2022; Cui et al., 2020; 2021). These models incorporate activation functions or other nonlinear structures, allowing them to better capture the brain's responses to complex stimuli and leading to improved predictive performance compared to linear models (Zhang et al., 2019; Li et al., 2022; Cui et al., 2020; 2021). However, as illustrated in Figure 1, nonlinear encoding models exhibit sample-specific characteristics, resulting in unstable structures that complicate the interpretation of underlying relationships.

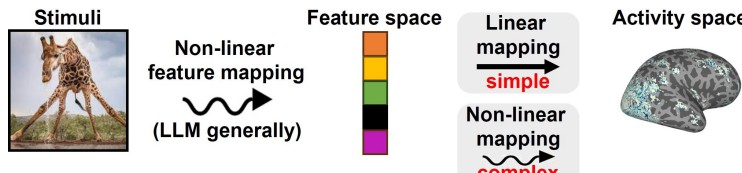

Figure 1: Comparison of linear and nonlinear encoding models. In linear encoding models, the mapping relationship between the feature space and brain activity space is invariant across input samples. On the contrary, nonlinear encoding models exhibit sample-specific characteristics, resulting in an unstable structure that complicates the interpretation of the underlying relationships.

In this study, we propose LinBridge, a learnable and flexible framework based on Jacobian analysis for interpreting nonlinear encoding models. The Jacobian matrix, which quantifies the output change rates relative to input, enables the analysis of sample-selective mapping in nonlinear models (Gale & Nikaido, 1965). LinBridge posits that the nonlinear mapping between the computational representations and neural responses can be factorized into two components: a linear inherent component that approximates the complex nonlinear dynamics, and a mapping bias that captures sample-selective nonlinearities. However, this factorization presents a dilemma: the sample-selective nonlinearities introduce substantial variability in the Jacobian matrices across different input-output pairs, complicating the extraction of consistent and interpretable mapping structures within nonlinear encoding models.

To address this challenge, we propose a self-supervised learning strategy based on contrastive learning, which has demonstrated superior capabilities in differentiating shared and distinctive attributes through paired-sample analysis (Oord et al., 2018; Schneider et al., 2023). Within the contrastive learning framework, LinBridge maximizes shared linear component within the Jacobian matrices while minimizing the influence of nonlinear features (nonlinear mapping biases) on those shared components, leading to an effective delineation of the linear inherent component and the nonlinear mapping biases. In addition, this self-supervised learning strategy allows LinBridge to adapt effectively to various nonlinear encoding models. Furthermore, LinBridge incorporates a low-dimensional embedding module that facilitates dimensionality reduction while preserving the intrinsic structure of the feature space, providing a more intuitive tool for analyzing the brain's linear and nonlinear responses to external stimuli. The contributions of this work are summarized as follows:

- We introduce LinBridge, a flexible framework designed to extract both the linear inherent component and the nonlinear mapping biases from nonlinear encoding models. This framework enables an interpretable analysis of the nonlinear mappings between computational representations and brain responses.

- The linear inherent component extracted by LinBridge exhibits activation patterns highly consistent with the original nonlinear encoding model, suggesting that the complex mappings in nonlinear neural encoding models can be effectively captured.

- We apply LinBridge to a neural encoding exploration of vision transformer models and reveal the variability in nonlinearity across different levels of the visual processing hierarchy.

## 2 BACKGROUND AND RELATED WORKS

### 2.1 LINEAR NEURAL ENCODING MODELS

Due to their simplicity and interpretability, linear neural encoding models are widely applied across various domains to disentangle the neural basis underpinning information processing in the brain, including vision (Yamins et al., 2014; Khaligh-Razavi & Kriegeskorte, 2014; Güçlü & van Gerven, 2015; Eickenberg et al., 2017; Zhuang et al., 2021), audition (Zhou et al., 2023; Li et al., 2023; MILLET et al., 2022; Tuckute et al., 2023; Vaidya et al., 2022), and language (Liu et al., 2023; Caucheteux & King, 2022; Goldstein et al., 2022; Jain & Huth, 2018; Schrimpf et al., 2021; Abdou, 2022). However, the inherent nonlinear dynamics of neural activity limit the predictive power and interpretability of linear models, particularly when addressing more complex cognitive func-

tions. This limitation is especially pronounced in higher-order cortical areas, which often involve complex and dynamic interactions, suggesting that their underlying neural mechanisms may not be adequately captured by linear representations (Naselaris et al., 2011b).

## 2.2 NONLINEAR NEURAL ENCODING MODELS

Nonlinear encoding models emerge as a solution to the limitations of linear models by incorporating nonlinear structures that more effectively capture the complex patterns of brain activity. These approaches demonstrate superior predictive performance compared to linear models (Zhang et al., 2019; Li et al., 2022; Cui et al., 2020; 2021). However, the opaque nature of nonlinear encoding models presents significant challenges in understanding the mapping between the computational representations and brain responses. To address this challenge, most existing methods adopt the framework proposed by (Tank et al., 2021), leveraging various time-series prediction models to enhance the interpretability of these learned black-box mappings (Khanna & Tan, 2019; Bussmann et al., 2021; Suryadi et al., 2023). In particular, the Jacobian matrix has been employed to elucidate the local mapping relationships in artificial neural networks (ANNs) (Zhou et al., 2024; Suryadi et al., 2023). Nonetheless, nonlinear models often exhibit sample-specific mappings, leading to substantial variability in the Jacobian matrices across different input-output pairs. This variability complicates the extraction of consistent and interpretable mapping structures within nonlinear encoding models.

## 3 METHODS

### 3.1 DATASET AND PREPROCESSING

We use the Natural Scenes Dataset (NSD) (Allen et al., 2021) in this study. The NSD dataset contains fMRI data from eight subjects passively viewing 73,000 color natural scene images over 40 hours. These images are cropped from the MS-COCO dataset (Lin et al., 2014), with each image displayed for three seconds and repeated three times across 30 to 40 scanning sessions, resulting in a total of 22,000 to 30,000 experimental trials. The fMRI data in the NSD are acquired using a whole-brain gradient-echo EPI (echo-planar imaging) sequence at 7T, with a resolution of 1.8 mm and a repetition time of 1.6 seconds. Single-trial beta maps are estimated using a customized general linear model and released alongside the raw fMRI data (Wang et al., 2022; Tang et al., 2024). Similar to previous studies (Wang et al., 2022; Tang et al., 2024), these beta maps are normalized (zero mean and unit variance) within each run and averaged across image repetitions to be used as functional brain activity measures. In our experiments, each image and its corresponding beta map are treated as a single sample. We divide the dataset into training, validation, and testing sets in a ratio of 8:1:1 (8000:1000:1000 samples). Notably, subjects 1, 2, 5, and 7 complete the full experimental protocol, and thus, their fMRI data are utilized in our experiments. In the experiments on neural encoding of vision transformer models, we use the pre-trained CLIP-ViT (Radford et al., 2021)[1] image encoder to derive the computational representation of the visual image stimuli.

### 3.2 ENCODING MODEL

The general encoding model can be formulated as follows:

$$\hat{y} = f(x) \tag{1}$$

where $x$ denotes the feature space of external stimuli, $\hat{y}$ is the brain activity space, and $f(\cdot)$ denotes the encoding function. Specifically, the computational representation of visual images spans the feature space, and the normalized beta maps serve as the brain activity space. To validate LinBridge, we construct both linear and nonlinear models in simplified form. Each model consists of two fully connected layers, with the nonlinear model additionally incorporating a ReLU activation function. Further details regarding the encoding models are provided in A.1. The mean squared error (MSE) (Huang et al., 2017) is used as the loss function in the encoding model.

---

[1]https://huggingface.co/openai/clip-vit-base-patch32.

### 3.3 COMPUTATION OF THE JACOBIAN MATRIX

The Jacobian matrix captures the local linear mapping between feature space and neural responses. It reflects the sensitivity of the neural encoder to variations in the input. Upon completion of model training, we fix the parameters of the encoding model and input the test set data to obtain the corresponding predictions. Given the $k$-th sample $x_k \in \mathbb{R}^d$ in the test set and the prediction of the nonlinear neural encoder $\hat{y}_k = f(x_k) \in \mathbb{R}^p$, the Jacobian matrix $\mathbf{JM}_k$ can be calculated by taking the derivative of the model's output with respect to its input as follows:

$$\mathbf{JM}_k = \left(\frac{\partial \hat{y}_k}{\partial x_k}\right)^T = \begin{bmatrix} \frac{\partial \hat{y}_{k,1}}{\partial x_{k,1}} & \cdots & \frac{\partial \hat{y}_{k,p}}{\partial x_{k,1}} \\ \vdots & \ddots & \vdots \\ \frac{\partial \hat{y}_{k,1}}{\partial x_{k,d}} & \cdots & \frac{\partial \hat{y}_{k,p}}{\partial x_{k,d}} \end{bmatrix} \in \mathbb{R}^{d \times 1 \times p} \tag{2}$$

We denote the collection of Jacobian matrices of all testing samples as $\mathbf{JM} \in \mathbb{R}^{d \times N \times p}$, where $d = 512$ is the dimensionality of the representation, $N = 1000$ is the sample size of the test set, and $p$ is the number of voxels.

### 3.4 LINBRIDGE

LinBridge leverages Jacobian matrices ($\mathbf{JM}$) to quantify the complex mapping relationships in nonlinear encoding models. It extracts the linear inherent component ($\mathbf{JM}_{\text{inherent}}$), which captures consistent, interpretable mapping structure invariant to input samples, and the nonlinear mapping biases ($\Delta \mathbf{JM}$) that reflect the unique nonlinear behaviors associated with distinct inputs.

#### 3.4.1 EXTRACTION OF THE LINEAR INHERENT COMPONENT

Similar to the dimensionality reduction architecture introduced in CEBRA (Schneider et al., 2023)[2], LinBridge employs a multi-layer convolutional neural network (CNN) to refine the input Jacobian matrices. Specifically, LinBridge progressively compresses the sample dimension of the Jacobian matrix using the CNN, enabling the extraction of the linear inherent component ($\mathbf{JM}_{\text{inherent}} \in \mathbb{R}^{d \times 1 \times p}$) that captures the linear mapping between the input representations of the nonlinear neural encoding model and the corresponding voxel activations in the brain. This approach provides a more structured interpretation of neural activity.

#### 3.4.2 NONLINEAR MAPPING BIASES $\Delta \mathbf{JM}$

We directly subtract $\mathbf{JM}_{\text{inherent}}$ from $\mathbf{JM}$ to obtain the nonlinear mapping biases $\Delta \mathbf{JM}$. This operation ensures that both the linear inherent component and the nonlinear mapping biases retain within the original space of $\mathbf{JM}$. Therefore, the computation of $\Delta \mathbf{JM}$ follows:

$$\Delta \mathbf{JM} = \mathbf{JM} - \mathbf{JM}_{\text{inherent}} \in \mathbb{R}^{d \times N \times p} \tag{3}$$

#### 3.4.3 LOW-DIMENSIONAL EMBEDDING

LinBridge incorporates a low-dimensional embedding module that linearly reduces the dimensions of $\mathbf{JM}_{\text{inherent}}$, $\mathbf{JM}$, and $\Delta \mathbf{JM}$ to a more compact representation. This linear dimensionality reduction effectively preserves the relative spatial distances of $\mathbf{JM}_{\text{inherent}}$ $\mathbf{JM}$, and $\Delta \mathbf{JM}$ in the latent space. On the one hand, lower dimensional features increase the effectiveness of contrastive learning, on the other hand, it reduces computational cost. Specifically, we use a fully connected (FC) layer to project each matrix into a low-dimensional space.

$$\mathbf{JM}_{\text{inherent}}^{\text{down}} = FC_{\text{Downsample}}(\mathbf{JM}_{\text{inherent}}) \in \mathbb{R}^{1 \times p} \tag{4}$$

$$\mathbf{JM}^{\text{down}} = FC_{\text{Downsample}}(\mathbf{JM}) \in \mathbb{R}^{N \times p} \tag{5}$$

$$\Delta \mathbf{JM}^{\text{down}} = FC_{\text{Downsample}}(\Delta \mathbf{JM}) \in \mathbb{R}^{N \times p} \tag{6}$$

In the equations above, the superscript "down" in $\mathbf{JM}_{\text{inherent}}^{\text{down}}$, $\mathbf{JM}^{\text{down}}$, and $\Delta \mathbf{JM}^{\text{down}}$ indicates the dimensionality-reduced representations. Generally, they are the result of applying FC module to the original $\mathbf{JM}_{\text{inherent}}$, $\mathbf{JM}$, and $\Delta \mathbf{JM}$, reducing them to a more compact, low-dimensional space.

---

[2]https://github.com/AdaptiveMotorControlLab/CEBRA.

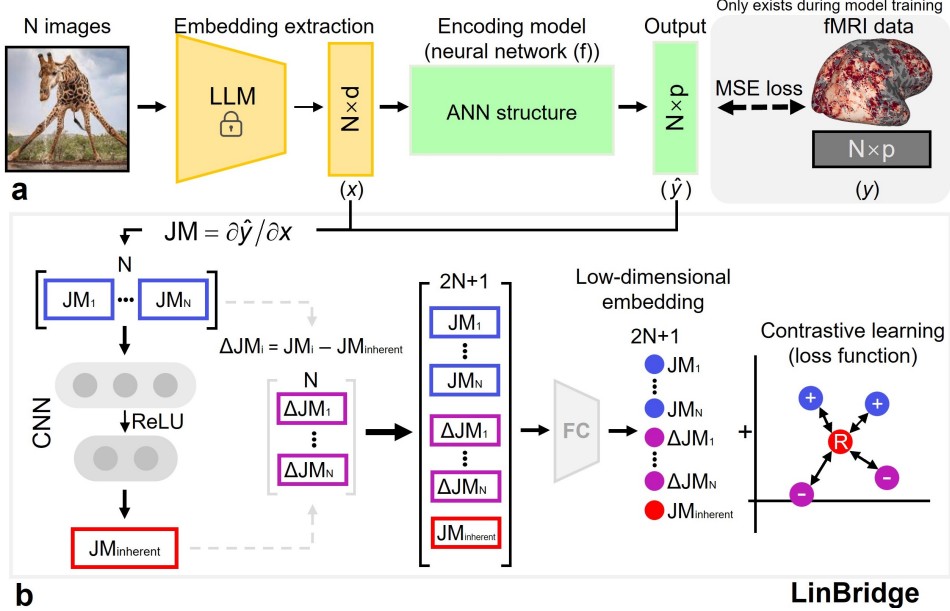

Figure 2: Nonlinear Encoding Model and the LinBridge Framework. (a) Image representation extraction and the general neural encoding model structure; (b) LinBridge framework, which includes the computation of $\mathbf{JM}$, the extraction of $\mathbf{JM}_{\text{inherent}}$ based on CNN module, the calculation of $\Delta\mathbf{JM}$, and the implementation of a low-dimensional embedding module.

### 3.4.4 Loss Function

The low-dimensional embeddings ($\mathbf{JM}_{\text{inherent}}^{\text{down}}$, $\mathbf{JM}^{\text{down}}$, and $\Delta\mathbf{JM}^{\text{down}}$) of these matrices are utilized for contrastive learning. Essentially, $\mathbf{JM}_{\text{inherent}}$ represents the shared linear components of $\mathbf{JM}$ in the latent space. The objective of the contrastive learning framework is to maximize the alignment of these components while minimizing the influence of the unique nonlinear characteristics represented by $\Delta\mathbf{JM}$. Consequently, it seeks to maximize the distinctiveness of $\mathbf{JM}_{\text{inherent}}$ in relation to $\Delta\mathbf{JM}$. To this end, LinBridge employs the InfoNCE loss function (Oord et al., 2018; Schneider et al., 2023)[3]. Contrastive learning maximizes the similarity between the reference sample and positive samples ($\mathbf{JM}_{\text{inherent}}^{\text{down}}$ and $\mathbf{JM}^{\text{down}}$), while concurrently minimizing the similarity between the reference sample and negative samples ($\mathbf{JM}_{\text{inherent}}^{\text{down}}$ and $\Delta\mathbf{JM}^{\text{down}}$). Thus, the InfoNCE loss function in this study is formulated as follows:

$$\mathcal{L}_{\text{InfoNCE}} = -\frac{1}{N}\sum_{i=1}^{N}\log\frac{\exp(\text{sim}(\mathbf{JM}_{\text{inherent}}^{\text{down}}, \mathbf{JM}_{i}^{\text{down}})/\tau)}{\exp(\text{sim}(\mathbf{JM}_{\text{inherent}}^{\text{down}}, \mathbf{JM}_{i}^{\text{down}})/\tau) + \sum_{j=1}^{N}\exp(\text{sim}(\mathbf{JM}_{\text{inherent}}^{\text{down}}, \Delta\mathbf{JM}_{j}^{\text{down}})/\tau)} \tag{7}$$

where $\text{sim}(\cdot, \cdot)$ denotes the similarity metric (e.g., cosine similarity), and $\tau$ is the temperature parameter. Additionally, we introduce L1-regularization to prevent overfitting of $\Delta\mathbf{JM}$:

$$\mathcal{L}_{\text{Reg}} = \lambda\sum|\Delta\mathbf{JM}| \tag{8}$$

where $\lambda = 0.01$ is the regularization coefficient. The final loss function is as follows:

$$\mathcal{L}_{\text{total}} = \mathcal{L}_{\text{InfoNCE}} + \mathcal{L}_{\text{Reg}} \tag{9}$$

In summary, the structural block diagram is shown in Figure 2. LinBridge utilizes a Jacobian matrix-driven strategy, enabling its application to any neural encoding model.

### 3.5 Evaluation Metrics

We use the coefficient of determination ($R^2$) to evaluate the predictive performance of the neural encoding models on the test set. To assess the statistical significance of the predictions, we follow

---

[3]https://github.com/RElbers/info-nce-pytorch.

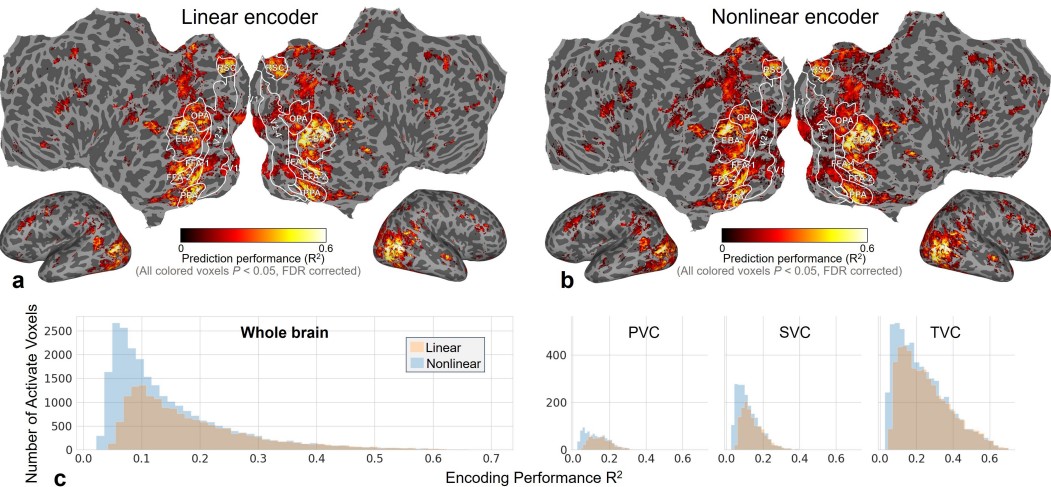

Figure 3: Comparison of $R^2$ between linear and nonlinear encoding models, showing predictions significantly above chance levels ($P < 0.05$, FDR corrected). (a) $R^2$ in the linear encoding model; (b) $R^2$ in the nonlinear encoding model; (c) The histograms of $R^2$ in the whole brain, the primary visual cortex (PVC), the secondary visual cortex (SVC), and the tertiary visual cortex (TVC). Results for other subjects are provided in A.3.

the method described in (Wang et al., 2023), conduct 200 bootstrapped resampling iterations on the test set and calculating FDR-corrected $P$-value thresholds for various performance metrics (Wang et al., 2023; Subramaniam et al., 2024).

## 4 RESULTS

### 4.1 PREDICTION OF VISUAL CORTEX WITH NONLINEAR ENCODERS

We compare the predictive performance of the linear and nonlinear encoding models. Figure 3 shows the $R^2$ values for both models for Subject 2 (all results reported in the main text are exemplified using Subject 2 unless otherwise stated). The results demonstrate that the nonlinear encoding model activates a broader range of brain regions compared to the linear model (24,490 voxels vs. 16,084 voxels), particularly in the visual cortex. Figure 3 (c) further illustrates that the nonlinear encoding model achieves significantly higher $R^2$ values across various visual areas, including the primary visual cortex (PVC: V1), secondary visual cortex (SVC: V2, V3, V4), and tertiary visual cortex (TVC: EBA, PPA, RSC, OPA, FFA-1, FFA-2). Notably, 9.30% of the voxels in the nonlinear encoding model exhibit relatively good predictive performance ($R^2 > 0.05$), in contrast to 6.65% in the linear model. These results suggest that the nonlinear encoding model outperforms the linear one.

### 4.2 LINEAR INHERENT COMPONENT: LINEAR INTERPRETATIONS OF NONLINEAR MAPPINGS

Figure 4 (a) illustrates the stability of the linear inherent component extracted by LinBridge. We calculate the Pearson correlation coefficients between the linear inherent component obtained at various batch sizes ($\mathbf{batchsize} \in [\mathbf{16}, \mathbf{32}, \mathbf{64}, \mathbf{128}, \mathbf{256}, \mathbf{512}]$) and the linear inherent component obtained at a batch size of 1000, that is, the entire test set. The training strategies are detailed in A.2. This analysis is repeated for 200 times to obtain means and standard deviations. As shown in Figure 4 (a), the inherent structure extracted by LinBridge is highly stable across all evaluated batch size, as evidenced by Pearson coefficients approaching 1. This high stability suggests that LinBridge can accurately capture the relationship carried by the encoding model even with small batch sizes.

The linear inherent component extracted by LinBridge closely aligns with the activation patterns in the nonlinear encoding model (Figures 3 (b) and 4 (b)). Additionally, we compare the $R^2$ values of the nonlinear encoding model and the linear inherent component extracted by LinBridge across

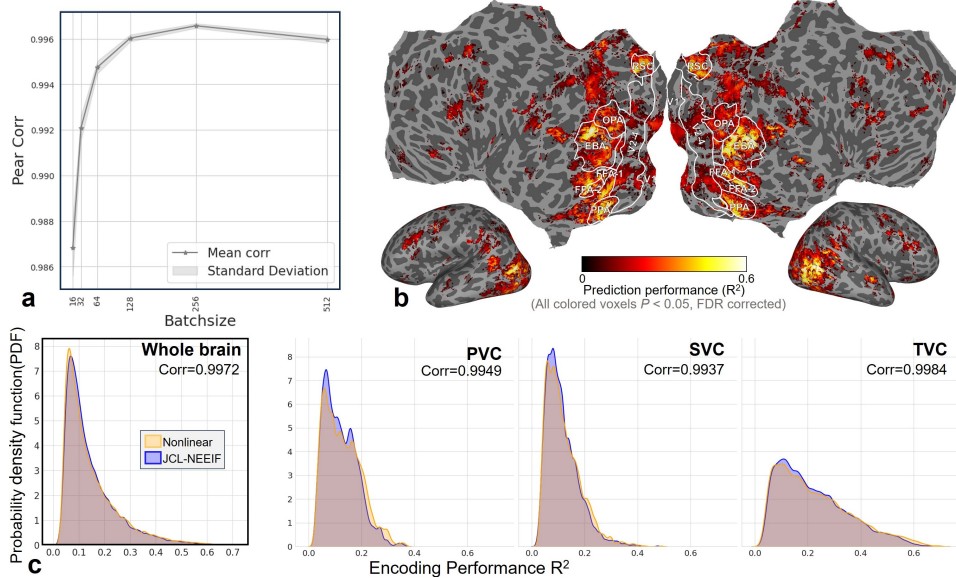

Figure 4: Comparison of the linear inherent component extracted by LinBridge to the brain activation predicted by the nonlinear encoding model. (a) The stability of the extracted linear inherent component across different batch sizes. (b) Activation patterns of the linear inherent component extracted by LinBridge. (c) Comparison of the distribution of $R^2$ values between the linear inherent component extracted by LinBridge and the original nonlinear encoding models in the whole brain, PVC, SVC and TVC. Results for other subjects can be found in A.4.

the whole brain, PVC, SVC, and TVC (Figure 4 (c)). The $R^2$ values in the two conditions exhibit high correlations, with Pearson correlation coefficients of 0.9972, 0.9949, 0.9937, and 0.9984 at the whole brain level, and in PVC, SVC, and TVC, respectively. The linear inherent component achieves comparable or even superior performance in specific brain regions. All these results demonstrate that the linear inherent component extracted by LinBridge can accurately capture the complex relationship represented by the nonlinear encoding model.

## 4.3 NONLINEAR ENCODING IN VISUAL CORTEX

The variations of $\Delta\mathbf{JM}$ regarding to various samples are valuable to characterize the nonlinearity of voxel-wise encoding model, and hence the visual cortex.

However, the high dimension of $\Delta\mathbf{JM}$ complicates the quantification of its variations among samples. In this context, the low-dimensional embedding $\Delta\mathbf{JM}^{\text{down}}$ derived from LinBridge provides a novel perspective for depicting the nonlinearity of the visual cortex. To this end, we incorporate linear fitting (Allen et al., 2021; Cohen, 1997; Hlinka et al., 2011) to assess how $\Delta\mathbf{JM}$ varies with different samples. Specifically, for each selected voxel, we first generate a sample index array and then extract the corresponding response values from $\Delta\mathbf{JM}^{\text{down}}$. We subsequently calculate the coefficients of a first-degree polynomial through polynomial fitting, with these coefficients representing the linear response weights of the voxel across different categories. It is hypothesized that a voxel responses to external stimuli linearly if the corresponding $\Delta\mathbf{JM}^{\text{down}}$ is invariant to different samples (i.e., the absolute value of the first derivative approaches 0). Conversely, the voxel is deemed nonlinear. Thus, we employ the absolute value of the first derivative (AFD) as a metric to evaluate the nonlinearity of a voxel.

We first perform dimensionality reduction for the Clip-Vit features of all the 73,000 stimulus images using t-SNE (Van der Maaten & Hinton, 2008). We then reorder the image samples in descending order according to their 1D-representation resulting from t-SNE. Figures 14-16 in A.5 show the sorted image samples, demonstrating an obvious transition from "simple" to "complex", as well as that the samples with similar semantics cluster together. This distribution pattern remains consistent across subjects.

Figure 5 (a) illustrates $\Delta\mathbf{JM}^{\text{down}}$ in the test samples for two voxels, in which the $x$-axis is the sorted image index according to their 1-D t-SNE representation in descending order. The left and right panels correspond to the voxels with the highest and lowest AFD values, respectively. In the left panel, the low-dimensional embedding of $\Delta\mathbf{JM}^{\text{down}}$ varies sharply with the sorted image samples, and the high AFD value indicates strong nonlinearity. Conversely, in the right panel, the low-dimensional embedding of $\Delta\mathbf{JM}^{\text{down}}$ is invariant to image samples, evidencing strong linearity of the voxel.

Figure 5 (b) shows the histograms of AFDs in PVC, SVC, and TVC. The number of voxels with relatively higher AFDs increases significantly from PVC to TVC. This observation may suggest a progressive nonlinearity within the hierarchy of visual cortex. Our findings closely align with previous research. For example, the primary visual cortex predominantly processes relatively simple visual features (Wang et al., 2023; Glasser et al., 2016; Huff et al., 2018), and consequently exhibits lower nonlinear encoding. In contrast, the middle and higher visual cortices manage more complex visual scenes (Wang et al., 2023; Glasser et al., 2016), such as spatial relationships and object recognition, exhibiting stronger nonlinear components in their neural activity.

Figure 5 (c) visualizes the nonlinearity measured by AFDs for significantly activated voxels across the whole brain. The PVC and SVC display more linear characteristics, whereas the TVC exhibits remarkable nonlinearity, further reinforcing the notion of hierarchical distribution of nonlinearity in the visual cortex. The probability density distribution of the AFDs in Figure 5 (d) indicates that the TVC demonstrates a higher probability of higher AFDs.

Further analysis reveals that, beyond the visual cortex, other brain regions associated with higher cognitive processing also exhibit nonlinearity (Figure 5 (c)), For instance, the temporoparietal-occipital junction (TPOJ) and prefrontal areas. The TPOJ, recognized as a key region for multimodal information integration (Wang et al., 2023; Glasser et al., 2016), exemplifies the complexity of non-linear processing and hierarchical information transmission in the brain. In parallel, the nonlinearity of the prefrontal cortex suggests that during higher cognitive tasks, such as decision-making and reasoning, the brain appears to rely on complex nonlinear mechanisms.

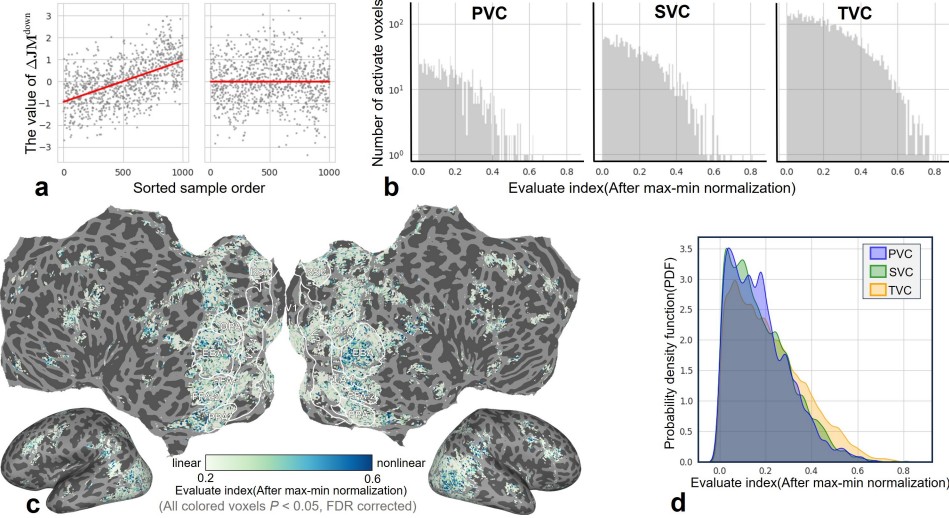

Figure 5: Distribution of nonlinear visual encoding in the brain. (a) The left and right images show voxel fitting results for the highest and lowest evaluation metrics, respectively; (b) Histogram comparison of evaluation metrics for significantly activated voxels in PVC, SVC, and TVC; (c) Visualization of evaluation metrics for significantly activated voxels across the whole brain; (d) Comparison of probability density functions for evaluation metrics of significantly activated voxels in PVC, SVC, and TVC. Results for other subjects can be found in A.6.

## 5 CONCLUSIONS, LIMITATIONS, AND PROSPECTS

In this study, we introduce LinBridge, a novel framework aimed at interpreting nonlinear neural encoding models through Jacobian analysis. We hypothesize that the intricate nonlinear mappings between ANN representations and neural responses can be decomposed into a linear inherent component and a sample-selective mapping bias. LinBridge effectively bridges the interpretability and nonlinearity divide, advancing our understanding of neural encoding and serving as a valuable tool for future investigations into complex neural computations across various brain regions.

The present study acknowledges several limitations. First, we validate LinBridge by using a simple nonlinear encoding model. In the future, it is interesting to conduct further validation studies using more advanced nonlinear encoding models. Second, while the Jacobian matrix elucidates the mapping relationships between samples and corresponding outputs, its computation remains resource-intensive when applied to large datasets. Given that pre-trained encoding models inherently contain gradient information, the incorporation of low-rank matrix decomposition for efficient computation of mapping relationships between samples represents a promising avenue for future work. Third, although our study focuses on the visual cortex, it is interesting to apply LinBridge to neural encoding models in other modalities such as acoustic, linguistic, and multimodal information.

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

# A APPENDIX

## A.1 NONLINEAR ENCODER AND LINEAR ENCODER

Figure 6: Structural comparison between nonlinear and linear encoding models.

To investigate the relationship between visual feature representations and brain responses, two ANN encoding models are constructed: a linear encoding model (Linear encoder) and a nonlinear encoding model (Nonlinear encoder).

**Nonlinear Encoding Model**

$$\hat{y}^{nonlinear} = W_2\sigma(W_1x + b_1) + b_2 \tag{10}$$

where $x$ denotes the input representations, and $W_1, b_1$ and $W_2, b_2$ are the weights and biases of the first and second layers, respectively. $\sigma$ denotes the ReLU activation function, which introduces nonlinearity to enhance the model's expressiveness.

**Linear Encoding Model**

$$\hat{y}^{linear} = W_2(W_1x + b_1) + b_2 \tag{11}$$

This model omits the ReLU activation function, allowing only linear transformations and serving to investigate the performance of linear encoding. In this study, the bias terms $b_1$ and $b_2$ are both set to False. Their structural comparison is illustrated in Figure 6. Meanwhile, the detailed code implementations of their models are presented in **Algorithms 1** and **2**, respectively.

**Training Strategy**

The training process of the encoding model utilizes the Adam optimizer (Zhang, 2018) for parameter optimization. To prevent overfitting, early stopping is employed, ceasing training if the validation loss fails to improve over 8 consecutive epochs. Additionally, the model parameters that demonstrate the optimal performance on the validation set are recorded and preserved.

---

**Algorithm 1** Nonlinear encoding model

---

```
class NonLinear_ANN_encoder(nn.Module):
    def __init__(self, input_dim=512, out_dim=p, hidden_dim=2048):
        super(NonLinear_ANN_encoder, self).__init__()
        self.encoder = nn.Sequential(
            nn.Linear(input_dim, hidden_dim, bias=False),
            nn.ReLU(),
            nn.Linear(hidden_dim, output_dim, bias=False)

    def forward(self, x):
        return self.encoder(x)
```

---

**Algorithm 2** Linear encoding model

```
class Linear_ANN_encoder(nn.Module):
    def __init__(self, input_dim=512, out_dim=p, hidden_dim=2048):
        super(Linear_ANN_encoder, self).__init__()
        self.encoder = nn.Sequential(
            nn.Linear(input_dim, hidden_dim, bias=False),
            nn.Linear(hidden_dim, output_dim, bias=False)

    def forward(self, x):
        return self.encoder(x)
```

## A.2  TRAINING STRATEGIES OF LINBRIDGE UNDER DIFFERENT BATCHSIZES

Here we demonstrate how LinBridge is trained with different batchsizes. Specifically, we provide the code implementation for LinBridge under various batchsize settings. Meanwhile, the detailed code implementations are presented in **Algorithms 3**. The code demonstrates how LinBridge is trained with different batchsizes, ranging from 16 to large 512 batchsizes.

**Algorithm 3** Training strategies of LinBridge under different batchsizes

```
JM = torch.load(project_dir)                                  ▷ Load Jacobian matrix data
batchsize_set = [16, 32, 64, 128, 256, 512]                          ▷ Set of batchsizes
for batchsize in batchsize_set do
    dataset = TensorDataset(JM)
    dataloader = DataLoader(dataset, batch_size=batchsize, shuffle=True, drop_last=True)
    info_nce_loss = InfoNCE(negative_mode='unpaired')
    optimizer = optim.AdamW(['params': LinBridge_model.parameters()], lr=1e-3)
    for epoch in 128 do
        for JM_batch in dataloader do
            optimizer.zero_grad()
            JM_inherent, delta_JM, JM_inherent_down, JM_down, delta_JM_down
                = LinBridge_model(JM_batch)
            loss = (info_nce_loss(JM_inherent_down.repeat(JM_batch.shape[1], 1),
                JM_down, delta_JM_down)
                + 0.01 * torch.sum(torch.abs(delta_JM)))
            loss.backward()
            optimizer.step()
        end for
    end for
end for
```

## A.3 COMPARISON OF $R^2$ BETWEEN LINEAR AND NONLINEAR ENCODING MODELS IN OTHER SUBJECTS

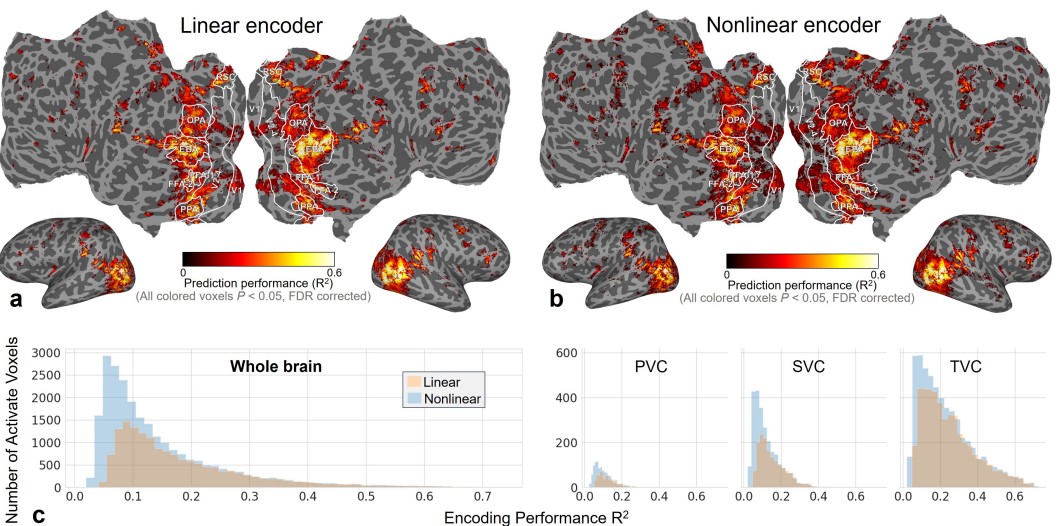

Figure 7: Comparison of $R^2$ between linear and nonlinear encoding models in Subject 1.

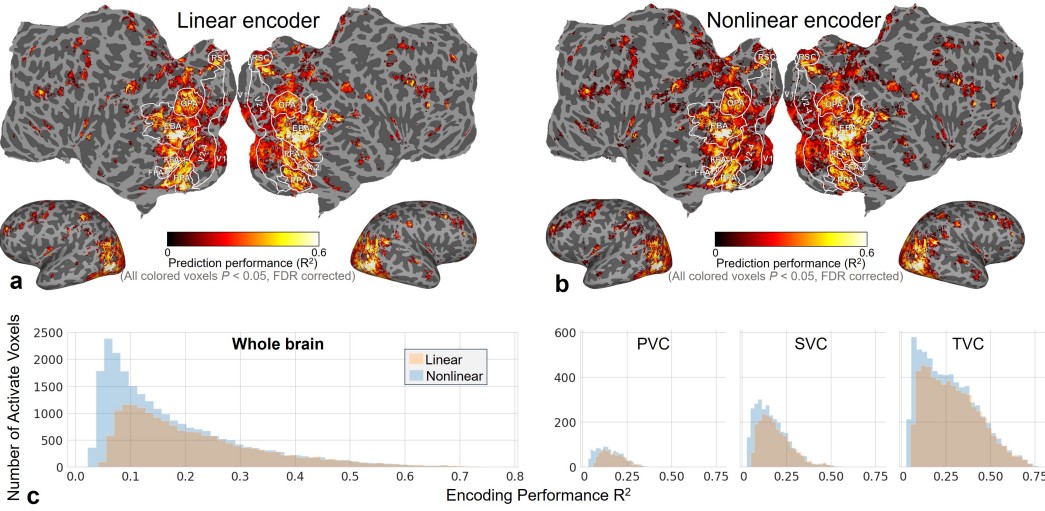

Figure 8: Comparison of $R^2$ between linear and nonlinear encoding models in Subject 1.

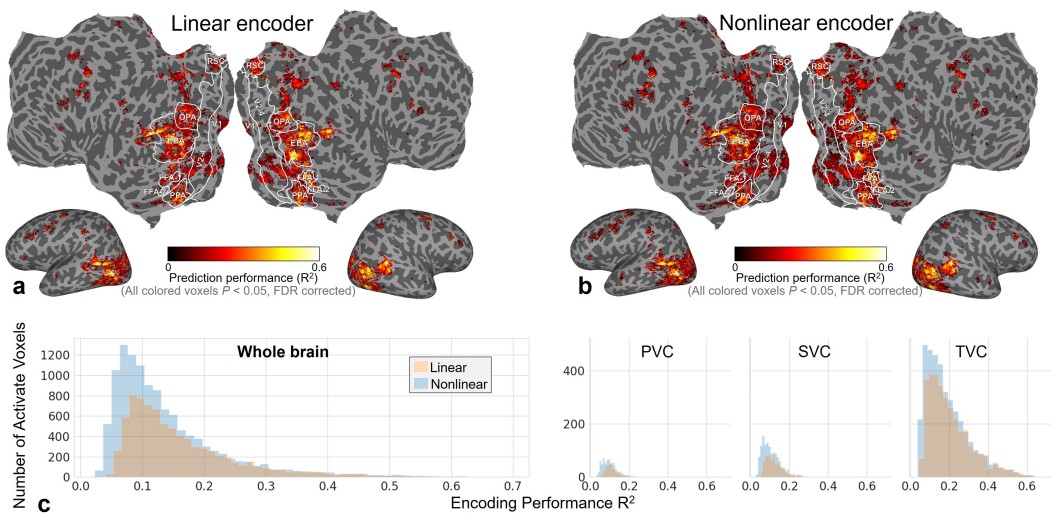

Figure 9: Comparison of $R^2$ between linear and nonlinear encoding models in Subject 1.

## A.4 COMPARISON OF THE LINEAR INHERENT COMPONENT EXTRACTED BY LINBRIDGE WITH BRAIN ACTIVATION OF THE NONLINEAR ENCODING MODEL IN OTHER SUBJECTS

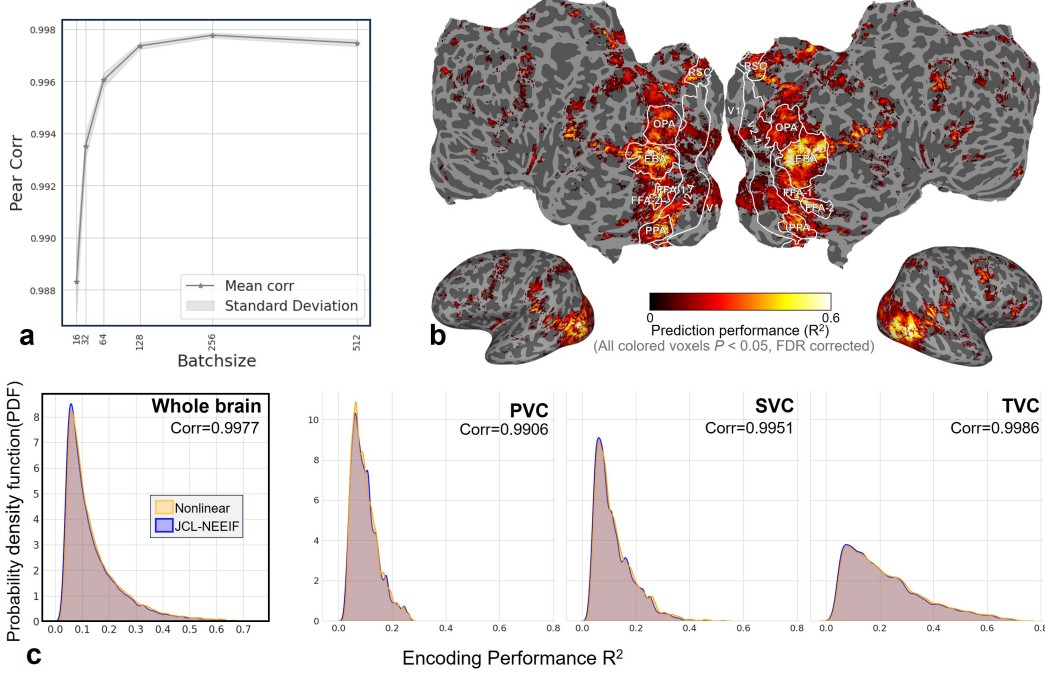

Figure 10: Comparison of the linear inherent component extracted by LinBridge with the brain activation of the nonlinear encoding model in Subject 1.

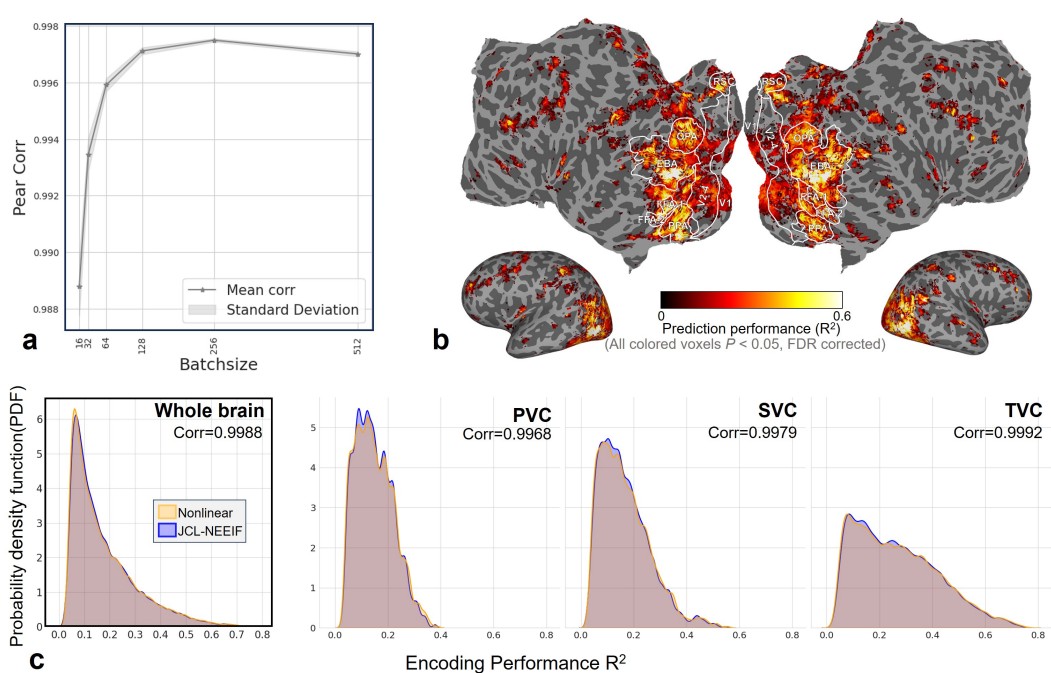

Figure 11: Comparison of the linear inherent component extracted by LinBridge with the brain activation of the nonlinear encoding model in Subject 5.

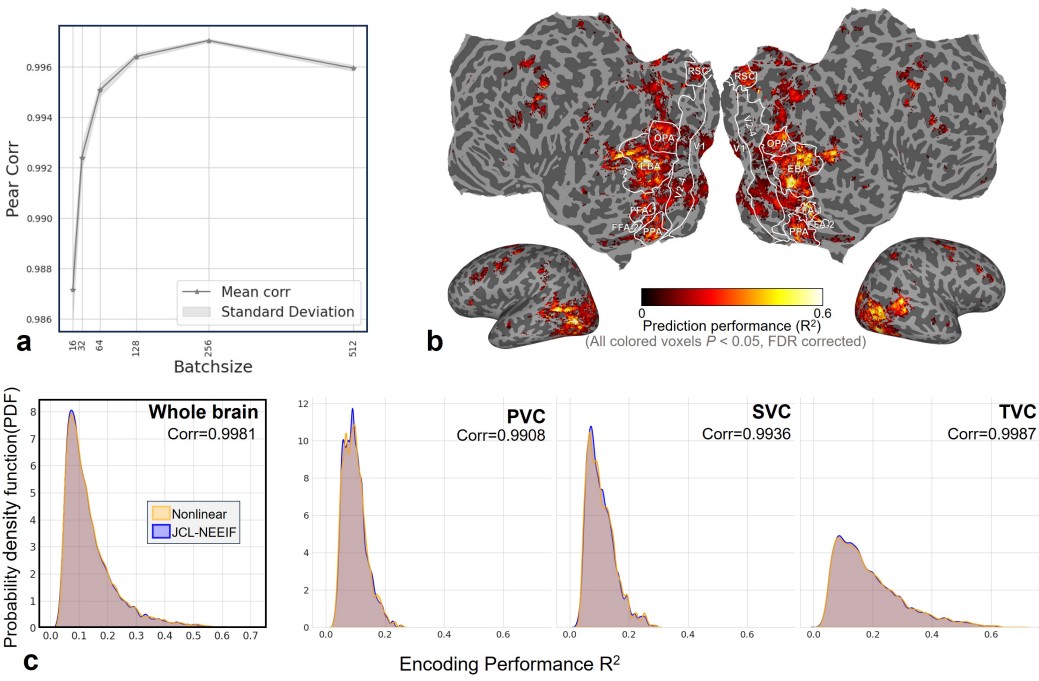

Figure 12: Comparison of the linear inherent component extracted by LinBridge with the brain activation of the nonlinear encoding model in Subject 7.

A.5   DISPLAY OF TEST IMAGES AFTER PROGRESSIVE SORTING

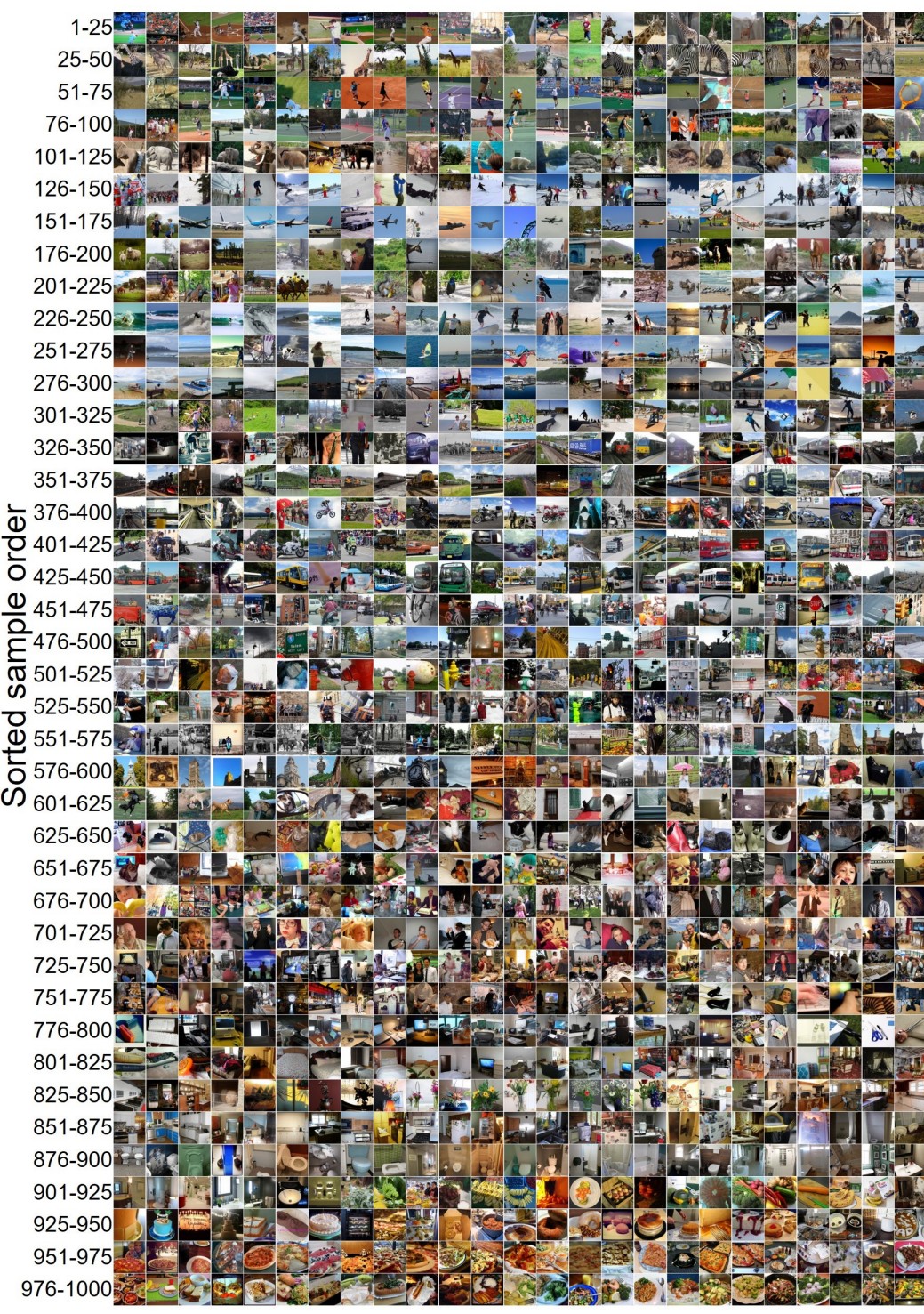

Figure 13: The images of the test set for Subject 2 after progressive sorting. Each image corresponds sequentially from top to bottom and left to right to the positions 1-1000 in the "Sorted Sample Order" of Figure 5 (a). From this, it can be observed that there is a certain pattern present in the progressively sorted test images.

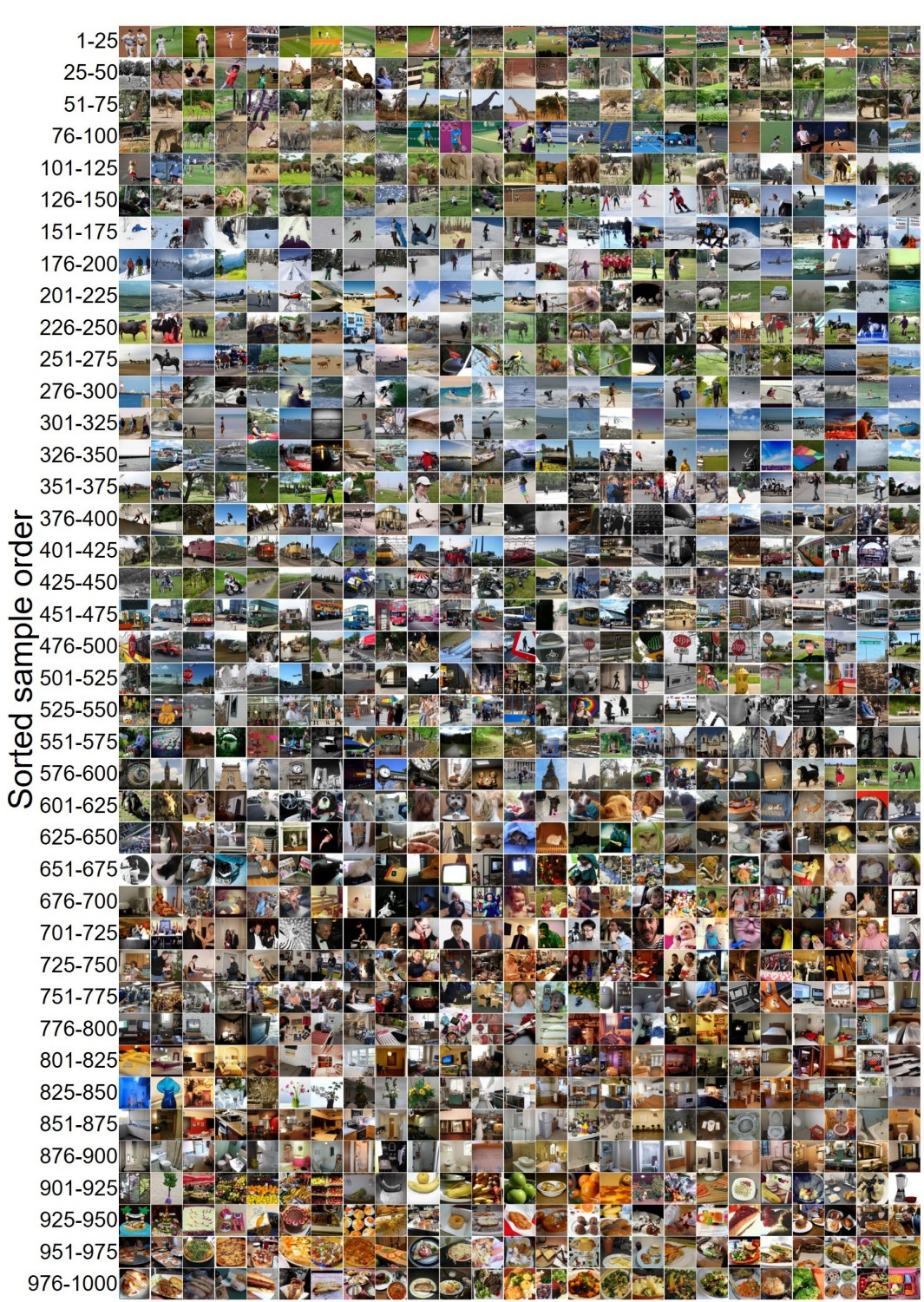

Figure 14: Display of test images for Subject 1 after progressive sorting.

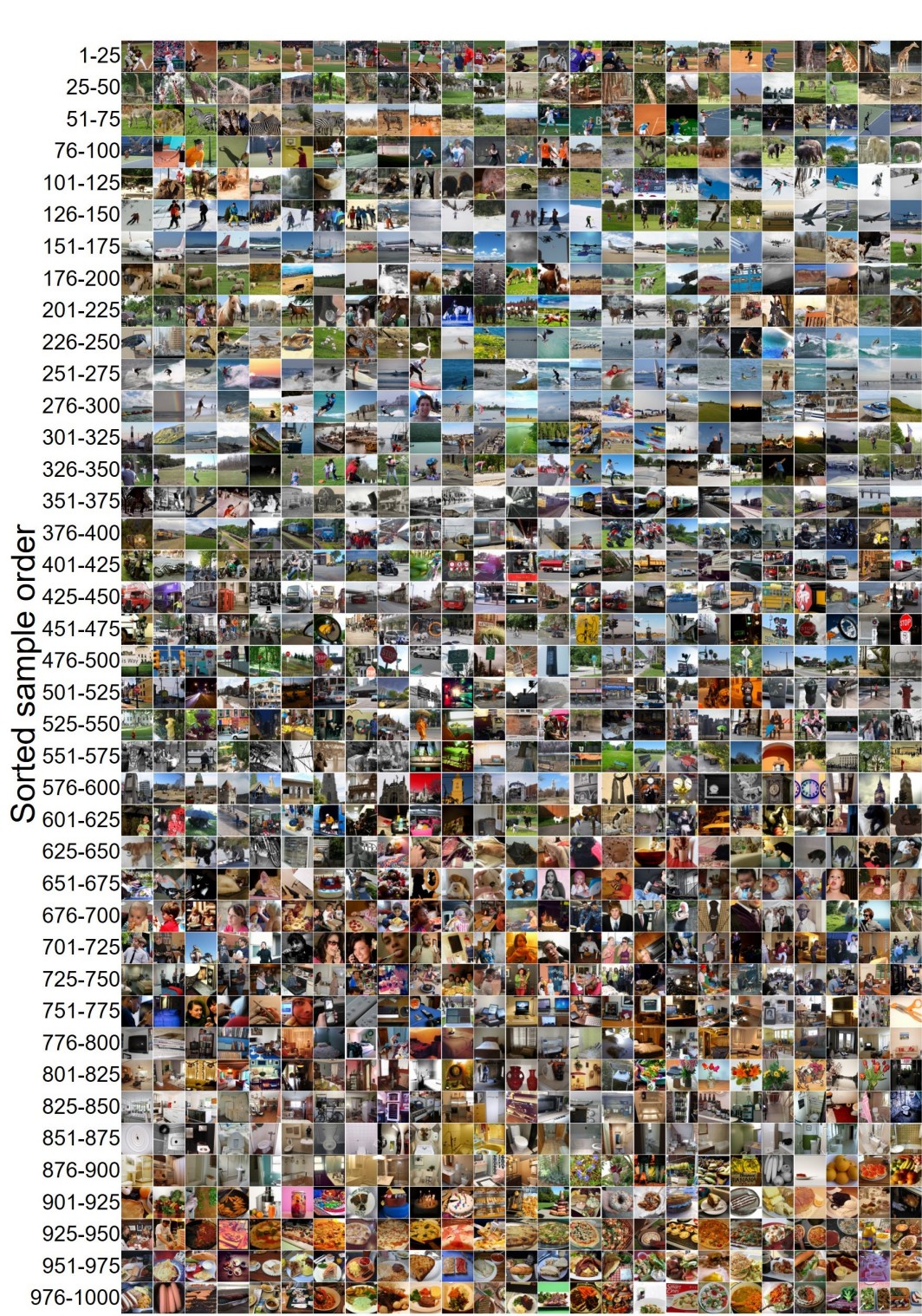

Figure 15: Display of test images for Subject 5 after progressive sorting.

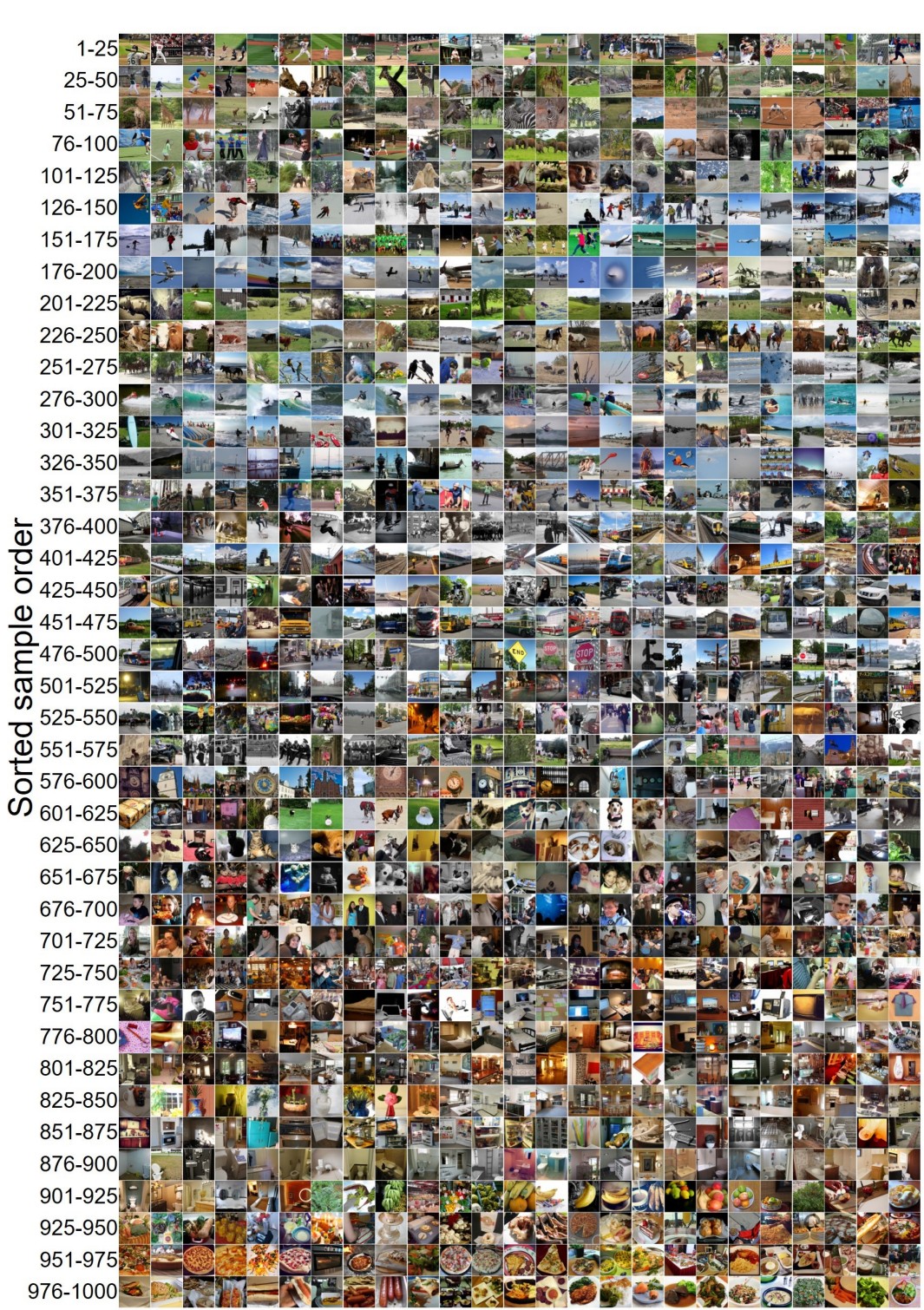

Figure 16: Display of test images for Subject 7 after progressive sorting.

## A.6 DISTRIBUTION OF NONLINEAR ENCODING IN THE BRAIN IN OTHER SUBJECTS

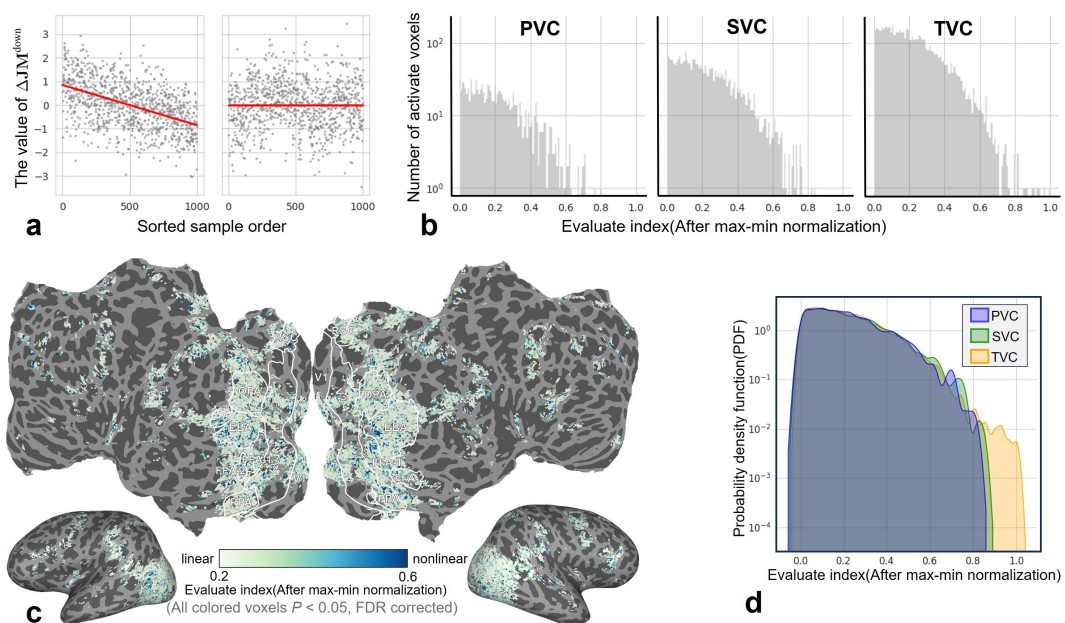

Figure 17: Distribution of nonlinear encoding in the Brain of Subject 1.

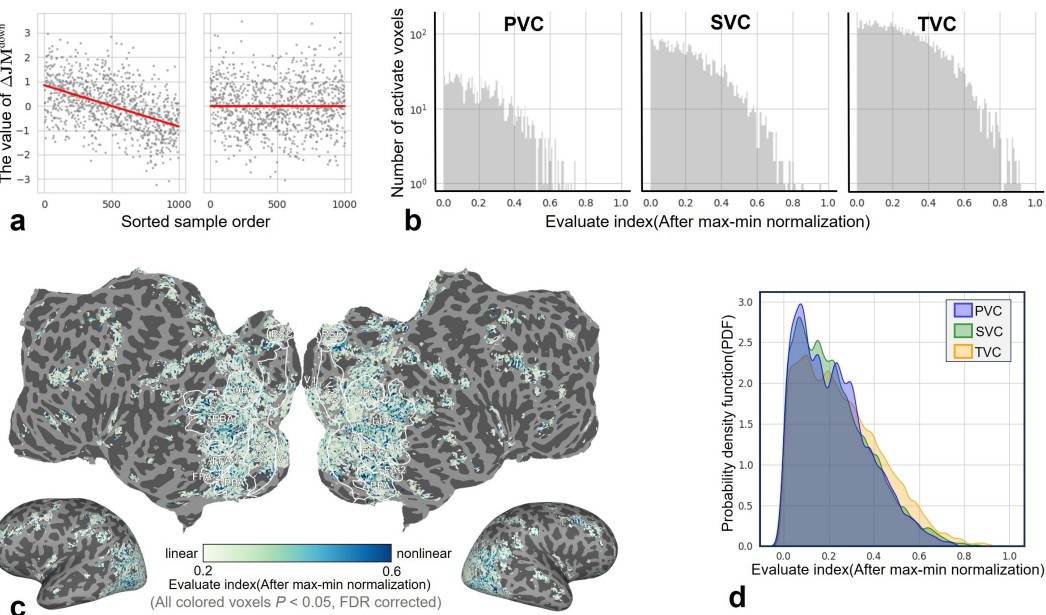

Figure 18: Distribution of nonlinear encoding in the Brain of Subject 5.

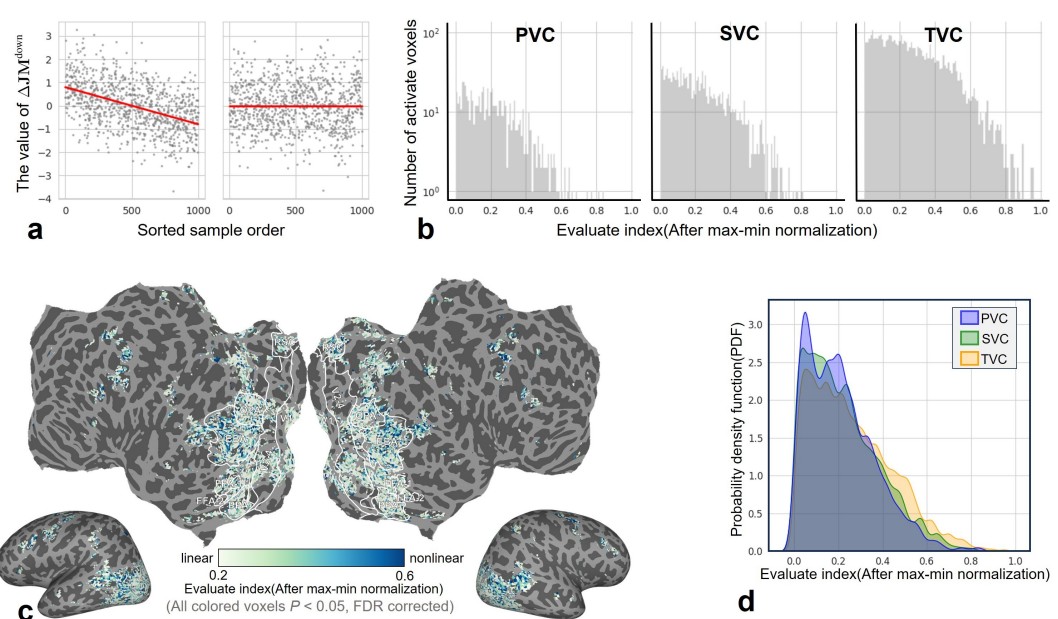

Figure 19: Distribution of nonlinear encoding in the Brain of Subject 7.

