# OpenReview forum: "LinBridge: A Learnable Framework for Interpreting Nonlinear Neural Encoding Models"
_ICLR.cc/2025/Conference — Submitted to ICLR 2025_

### Official Review · Reviewer_rUdW · 2024-11-02

**Soundness:** 2
**Presentation:** 2
**Contribution:** 2
**Rating:** 3
**Confidence:** 3

**Summary:**

The authors presents a novel encoding model architecture designed to disentangle linear from nonlinear relationships between visual representations and neural responses. The authors demonstrate the framework's utility through its application to fMRI data an successfully reveal hierarchical patterns of nonlinearity in the cortical visual processing stream.

**Strengths:**

The authors present an overall very clearly written paper. The problem statement, the approach and the performed analysis are well described. The approach is promising, but is severely limited by weak evaluations.

**Weaknesses:**

Despite the clear presentation and interesting ideas, the paper is severely limited by the weak evaluations and missing baselines. The model performances of both the linear and non-linear model are not only very similar, but seem to be fairly low overall, casting doubt on the validity of the approach.
Considered as a whole, there needs to be a substantial revision that addresses these weaknesses, which would improve this paper significantly.

*Major Concerns*:
- I'm not convinced of the validity of the overall approach. The authors claim that their approach allows disentangle the contributions of linear and nonlinear processing in the visual hierarchy. For this, they use a highly nonlinear feature extractor (CLIP-ViT) and then combine these features either linearly or non-linearly to predict responses in visual cortex. This method is severely limited in its explanatory power because it so crucially depends on the chosen non-linear feature extractor. A large model like a ViT very likely has an overcomplete basis, such that the differences in linearly or non-linearly combining these features will be very small. There is evidence for this in the nearly identical evaluation metrics between these two models.

- The model performance overall seems very week, by visualizing the voxel-wise predictive performance, it seems that many voxels can't be more accurately predicted as compared to chance level. On fMRI data of the visual cortex, relatively simple LNP-models usually perform better than chance level.

- Because the models are so similar, it is crucial to compare that against a baseline with a different architecture and a different training paradigm to distinguish the contrastive training from a simple regression-based neuronal response fitting.

- The approach could further be studied more comprehensively by validating it on toy-data, such as combinations of gabor-like V1 simple and complex cells and non-linear combinations of these. There, it would be expected that LinBridge is able to disentangle the linear from non-linear components.


*Minor Concerns*:
- the authors write in the introduction: "We apply LinBridge to a neural encoding exploration of vision transformer models and reveal the variability in nonlinearity across different levels of the visual processing hierarchy". This does make it sound as if the approach was used to study intermediate representations of vision transformer models. However, such as analysis was not performed and is misleading.

**Questions:**

- The description of the encoding model is not complete. Where the same intermediate features of a CLIP-ViT used to predict all voxels?

- in Fig. 2, the embedding extraction is denoted as "LLM" in the graphic. Is this supposed to represent the CLIP model?

- It is not clear why the contrastive training is needed. Could the authors provide further motivations why this strategy is meaningful here?

---

### Official Review · Reviewer_FTUv · 2024-11-03

**Soundness:** 3
**Presentation:** 2
**Contribution:** 3
**Rating:** 8
**Confidence:** 4

**Summary:**

The manuscript proposes a method called LinBridge to interpret non-linear encoding models of brain activity (models that predict the output response, given an input stimulus), by distinguishing the global Jacobian matrix (the linear part of the model) and local deviations from it around each single input presented in an experiment (the ‘biases’ that characterize the nonlinearity). Given a pre-trained encoding model, LinBridge uses contrastive learning on low-dimensional embedding of the Jacobian matrices. The manuscript demonstrates the method by first training a simple nonlinear model (a 2 layer network with sigmoid nonlinearity, and also a control linear model), on a large fMRI dataset of brain responses to images; and then applying LinBridge to the trained model.

The main results are 1) The non-linear model, despite its simplicity, predicts neural activity better than the linear model. 2) In the non-linear model, the linear global component characterized by LinBridge is almost as predictive as the full model. 3) The local ‘biases’ add more predictive power for higher visual cortex than lower visual cortex, consistent with general knowledge that higher visual cortex computes a more nonlinear representation of the image.

**Strengths:**

The problem addressed here is important. Learned nonlinear encoding models are necessary to predict brain activity well, but they are difficult to interpret and therefore offer limited understanding of brain computation. The proposed solution is conceptually simple and, to the extent tested in this manuscript, effective. The method combines well established ideas including the Jacobian to characterize local linear behavior, low-dimensional embeddings, and contrastive learning. The pseudo-code in the Appendix also highlights the simplicity of the method. With some exceptions noted below, the manuscript is written clearly.

**Weaknesses:**

The nonlinear encoding model is very simple and, not surprisingly, its predictive power is relatively small (how does it compare to other, previously published non-linear models for this dataset?). Although maximizing predictive power is not the goal of this manuscript, if the nonlinearity of the model is too simple, it seems not that surprising that the simple decomposition of LinBridge will work well.

In other words, how much does the nonlinearity really help this particular encoding model? Figure 3 is confusing in this regard: panels a,b look indistinguishable to me (sorry, I am not that familiar with fMRI data). And panel c shows that more voxels are activated for nonlinear, but the average r-square seems even lower for nonlinear than linear (linear histograms are shifted to the right). I suspect I am not reading the plots the right way, some guidance would help the reader.
Then figure 4 shows that the encoding performance with the full nonlinear model is almost identical to the linear component extracted by LinBridge. Doesn’t this say that the nonlinear part of the model is not that useful? The analysis of Figure 5 partly addresses that concern, showing that the nonlinear biases are more predictive in higher visual cortex.

Clarity: the computation of the Jacobian matrix JM should be explained clearly (not the linear and Delta terms, those are clear enough). It is the central idea of the paper, and the authors also list as a limitation that it is resource intensive, but it is not explained at all. It is conceptually simple, but I think the practical details might be important.
I found Figure 1 not informative. If the point is to convey that nonlinear mappings are characterized by the fact that they vary around different inputs, that is not at all conveyed by the image.

**Questions:**

See specific questions in weaknesses, in particular referring to Fig. 3-4, and to the details of computing the Jacobian.

---

### Official Review · Reviewer_n9vK · 2024-11-03

**Soundness:** 2
**Presentation:** 1
**Contribution:** 2
**Rating:** 5
**Confidence:** 2

**Summary:**

This paper seeks to interpret non-linear encoding models, showing that the "non-linearity" of the mapping CLIP-ViT representations to NSD responses in the brain increases with the brain regions' ranks in their hierarchy.

Disclaimer: I didn't understand the central use of J_inherent. I could not understand what exactly the authors mean by it - a formal definition is absent; Hereforth, I'm intuiting that it is somehow a measure of linearness in the mapping.

**Strengths:**

Intepreting non-linear encoding models is critical because they do tend to outperform linear encoding models as the paper shows. This paper seems to take a step in that direction.

**Weaknesses:**

As I mentioned before, the most critical element J_inherent isn't formally defined - making it very hard for me to understand what's happening next (although I can follow the intuitions).

**Questions:**

1. As far as I can tell, the non-linear encoding model doesn't do much better than the linear one in terms of R^2. It does predict more voxels though. Is the fact that you see more "non-linearness" in the mapping in later brain regions of the hierarchy due to the fact that the linear model cannot predict many of those voxels? Put another way, could a simple comparison between the R^2 given by a linear model vs the non-linear model, voxel-by-voxel, reveal similar profiles i.e. linear model does worse in certain brain areas and therefore the mapping needs to be non-linear?
2. What does the mapping being non-linear entail? You are using CLIP-ViT to encode responses of voxels across a huge chunk of the brain. Is it expected that a linear map should be possible in the first place? The brain presumably does way more things than just encode visuo-semantics the way CLIP-ViT does. I would've predicted that the mapping would be more linear in later visual areas as they are more semantic - turns out that's not the case. What gives, according to you? Q1 probably gives hints - does the linear model do much worse in high-level visual cortex? If yes, then possibly you're onto something super interesting.

---

### Official Review · Reviewer_9nLx · 2024-11-04

**Soundness:** 2
**Presentation:** 2
**Contribution:** 2
**Rating:** 3
**Confidence:** 4

**Summary:**

The paper presents a LinBridge, a flexible framework to extract both the linear inherent component and the nonlinear mapping biases from nonlinear encoding models. However, the performance of LinBridge in terms of fMRI prediction and interpretable visual feature extraction should be fully validated. And the hierarchical nonlinearity within the visual cortex is not convincing.

**Strengths:**

Using Jacobian matrices (JM) to quantify the complex mapping relationships in non-linear encoding models is interesting. The JM_{inherent} and \Delta JM is conceptually reasonable.

**Weaknesses:**

1. The LinBridge relies on an assumption that the nonlinear mapping between ANN representations and neural responses can be factorized into a linear inherent component that approximates the complex nonlinear relationship, and a mapping bias that captures sample-selective nonlinearity. This assumption is rather strong. Any evidence?
2. The computational cost of Jacobian is not reported.
3. Experiments are not rich. More model comparisons are needed.
4. This paper claims that the variability of nonlinearity across different levels of the visual processing hierarchy. However, the evidence of the distribution of hierarchical nonlinearity within the visual cortex is rather weak and highly dependent on the nonlinear embedding. The method should be fully validated before applying it to neuroscience discoveries. Moreover, evidence from previous neuroscience studies/experiments is needed to support this statement.

**Questions:**

See previous section.

Additional questions:
1. Figure 1 does not effectively illustrate the sample-specific characteristics and structural instability of the nonlinear encoding models, despite the authors' claims.
2. The method is not clearly described. For example, LLM in Figure 2 has not been defined in the Method. Is it CLIP-ViT? CLIP-ViT is not LLM model, but multi-modal visual-language model.
3. Figure 3 compared LinBridge nonlinear encoder with linear encoder in terms of fMRI prediction R^2. It is a non-fair comparison since nonlinear model has better fitting capacity. I would suggest comparing LinBridge with another nonlinear model with contrastive learning, for example, CEBRA, as well as the model in [1].
4. What are the features corresponding to the linear responses? Would the features change between using the linear model and LinBridge model?
5. What are sample-selective nonlinear responses? Can you please visualize it?

Ref
[1] Wang, A. Y., Kay, K., Naselaris, T., Tarr, M. J., & Wehbe, L. (2023). Better models of human high-level visual cortex emerge from natural language supervision with a large and diverse dataset. Nature Machine Intelligence, 5(12), 1415-1426.

---

### Official Review · Reviewer_dkMG · 2024-11-10

**Soundness:** 3
**Presentation:** 2
**Contribution:** 2
**Rating:** 3
**Confidence:** 4

**Summary:**

The authors introduce LinBridge, a framework for non-linear mapping between neural network activations and fMRI data (here, the Natural Scenes Dataset), with the goal of enhancing interpretability by factorizing the mapping into (i) a linear component that approximates the nonlinear relationship between source and target, along with (ii) a bias that captures the idiosyncratic aspects of the mapping that are "sample-specific" (given that in a non-linear encoding model, distinct samples may be transformed differently depending on their features). The authors motivate the need for such an approach by pointing out that the brain computes complex non-linear functions of the input, and thus, it is likely the case that non-linear mappings could provide better predictivity than linear mappings.

LinBridge works by leveraging Jacobian matrices, which capture the model's sensitivity to input variations. It separates the consistent linear component, JMinherent, from sample-specific nonlinear biases, ΔJM, using a CNN to compress the Jacobian matrix and extract meaningful structure. LinBridge then applies contrastive learning with InfoNCE loss to maximize alignment between consistent mappings and minimize nonlinear bias effects.

Using this approach, the authors show that that a nonlinear encoding model applied to NSD yield a greater percentage of voxels with significant encoding levels than a linear encoding model with the same 2-layer architecture, but no relu non-linearity. LinBridge successfully extracts a stable linear component from nonlinear models, which achieves comparable performance across visual regions.

Additionally, the authors use this framework to assess the degree to which a given voxel responds linearly with respect to the input samples.  They do this by measuring how the low-dimensional embedding ΔJMdown varies across different input samples. Then for each voxel, a first-degree polynomial is fit to the response values from ΔJMdown across an array of ordered samples. The coefficient of the first derivative from this linear fit reflects how much the voxel's response changes with different inputs. If the voxel's responses show minimal variation (i.e., the absolute first derivative - AFD - approaches zero), it indicates a more linear response; higher values imply stronger nonlinearity. Using this AFD metric, the authors present evidence that a greater proportion of visual voxel responses are non-linear along the visual hierarchy, from primary visual cortex to high-level visual cortex where category-selective regions are located.

**Strengths:**

- The authors take seriously the need to balance having mappings that are maximally predictive, and also interpretable. These two desiderata are naturally in tension, so I applaud that they developed a novel approach for tackling this challenge that yielded some sensible results.
- The authors do a solid job citing recent literature from the encoding model/mapping literature, with only a few key omissions. The work is certainly timely.
- I applaud the authors' use of NSD in this study - it's the data resource most naturally suited to developing new mapping approaches like these, and is substantially higher quality and larger scale than the datasets that were used in the papers by Zhang, Li, and Cui in section 2.2.
- The comparison between the 2-layer linear and non-linear models is a nice clear setup for validating their proposed approach.
- Using the factorization of the Jacobian to study to which voxels' responses are linear vs. nonlinear (using the AFD metric) is the most interesting part of the paper in my opinion. It's a clever application of their framework that could be more broadly applicable. I would like to see the AFD analyses extended in a revised version of this work.

**Weaknesses:**

I had several concerns with the paper's introduction and framing, which I will describe below. Beyond the framing of the paper and the way the authors discuss linear vs. nonlinear mappings, my main critique is that the validation and application of LinBridge did not provide much new scientific insight. I feel that the approach is mostly sound, but the impact of the paper is strongly limited by the fact that the authors do not do enough to show that LinBridge can provide new scientific insight into the nature of brain representations, to held build better theories. Relatedly, it also appears that the overall complexity and computational requirement of the approach is dramatically higher than linear encoding procedures. Based on the authors' results, it remains unclear to me whether (a) the added complexity of the procedure provides enough added predictive power to justify using LinBridge, and (b) whether factorizing the linear and non-linear components of the mapping actually provides the sort of "interpretability" that we seek in the field today. Folks who study explainable AI / mech-interp would use this phrase to refer more to understanding the nature of the features that are represented in DNNs and in the brain. The authors did not convince me that LinBridge gets us closer to this goal. For these reasons, I am unable to rate the paper higher than a 3, though I would be willing to revise my score upward with sufficient revision.

Comments on framing:

- The authors missed some important literature on linear vs. nonlinear mappings, such as [Ivanova et al. 2022](https://www.biorxiv.org/content/10.1101/2021.04.02.438248v3). This line of work should be cited. There's also an important recent [review](https://arxiv.org/abs/2310.13018) of mapping methods in neuroAI that the authors should cite. Their approach and findings should be meaningfully situated in the context of these two papers, since these more closely reflect the current state of thinking in neuroAI around this topic.

- The logic in Section 2 was a bit clunky. The authors critique linear encoding models by saying "the inherent nonlinear dynamics of neural activity limit the predictive power and interpretability of linear models". And later: "especially ... in higher-order cortical areas, [the] underlying neural mechanisms may not be adequately captured by linear representations". But, the linear encoding papers they cite nearly always fit the weights on top of deep and highly nonlinear DNN backbones. In other words, the authors misrepresent the typical use of linear encoding, giving the impression that the entire mapping function is linear. The more targeted and appropriate way to introduce this dichotomy would be: assuming some frozen feature extractor backbone, the trainable weights that map to brain data can either be an additional linear layer or an additional multi-layer function with some non-linearity.

- Another comment on section 2: the evidence the authors cite in favor of nonlinear models is insufficient to justify the blanket claim of "superior performance" compared to linear models. Beyond the Zhang, Li, and Cui papers, are there any that use more recent datasets (especially NSD) that directly compare nonlinear and linear mapping functions, and show the superiority of the former?

- Relatedly, because nonlinear functions are so much more expressive, they require much more data to fit, and often, stronger forms of regularization. For many datasets in the field, linear models may actually provide superior performance (in terms of generalization to held-out data), because they are less prone to overfitting. NSD is a great dataset for the purposes of studying nonlinear mappings, but the conclusions derived from NSD may not generalize to scenarios where fewer datapoints are available for fitting complex nonlinear functions.  This is one of the many reasons that the appropriate mapping method necessarily depends on one's research goals (see Ivanova et al. 2022) and also on the specifics of one's dataset - the blanket claim that non-linear models are inherently superior because of their computational expressivity is not clearly justified given the limitations inherent in most neuro datasets.

Comments on clarity, figures, results:

- The "sample-specific" idea was described in a vague way in the abstract and introduction - the authors should more clearly convey what this means. If I understand correctly, it refers to the following idea: say you have a vector of DNN activations in response to an image. No matter what that vector contains, the linear weights will apply the same transformation to it. However, in the presence of a nonlinearity and a multi-stage nonlinear mapping, the nature of the transformation post-nonlinearity will depend on the specific content of the original input vector. The logic here should be spelled out more clearly.

- The authors repeatedly refer to an "LLM" in their figure schematics, but the input is always an image, and the brain data they are using is visual. This could be a typo?

- The figure 2 caption did not do an adequate job explaining the complex methods schematic.

- Figure 3 shows that a greater proportion of voxels achieve p > 0.05 encoding, but that threshold appears to be quite low - the gains from LinBridge over a linear model seem to mostly involve the voxels with weakest signal. A positive interpretation is that the non-linear function somehow has a denoising effect, allowing us to predict some bits of cortex that were too noisy to be fit with a linear model. But, I am troubled by an alternative interpretation: what if these voxels with significant predictivity at the left tail of the blue distributions in Figure 3 arise because the nonlinearity is essentially just modeling noise structure in the data? With only 3 trials available, averaging will be insufficient to fully eliminate noise, and sources of signal and noise variability are possibly correlated in NSD and many other datasets. Is there any analysis that would convince the reader that these voxels with R2 < 0.1 but significant predictivity actually carry useful signal that can teach us something new about visual representation?  On this topic, I feel it would be more interesting if the LinBridge approach conferred better predictivity in the right tail of the distribution, suggesting that even in high-SNR voxels, there are non-linear components that linear models cannot explain. But, this does not seem to be the case based on Figure 3.  At minimum, the authors should provide a graph that quantifies the performance of linear vs non-linear mappings across a range of R2 thresholds spanning from 0.05 to 0.6 or so.

- While the results presented in Figure 5 seem to align with past work and suggest that higher-level voxels show more nonlinearity in their responses than earlier voxels, I wonder if this is more of a sanity check than a relevant scientific finding? Wouldn't the null hypothesis be exactly this, under the assumption that feedforward visual processing implements successive nonlinear transformations of the representations that begin in V1?  The authors could do more to convey the importance of this finding - perhaps I am not fully understanding the implications.

**Questions:**

I have described my questions and suggestions above in the Weaknesses section. Beyond merely validating that the assumptions of LinBridge hold and that the technique slightly raises R2 scores across some parts of cortex compared to linear mappings, I would strongly recommend that the authors perform additional analysis to convey that LinBridge can provide more rich and detailed forms of insight into visual representation, over and above a linear encoding model applied to the same dataset.

---

### Meta-Review · Area_Chair_Tign · 2024-12-22

**Metareview:**

This paper introduces a novel framework for characterizing and interpreting the relationships between artificial neural networks and fMRI imaging data.  The reviewers praised the paper for proposing a conceptually simple and effective solution to a timely and important problem. Unfortunately, however, they were not persuaded that it provided a rigorous or thorough enough set of evaluations and comparisons to existing baselines, and raised concerns about the reliance on a pre-specified nonlinear feature extractor and the ability to provide new scientific insights.    I regret that the paper cannot be accepted to this year's ICLR, but I wish the authors the best of luck in revising it for publication elsewhere.

**Additional Comments On Reviewer Discussion:**

The reviewers raised a number of concerns, including about the thoroughness of comparisons to existing baselines and the paper's ability to provide new scientific insights.  The authors did not write any rebuttals.

---

### Decision · Program_Chairs · 2025-01-22

Reject